



# Coupling wildfire spread simulations and connectivity analysis for hazard assessment: a case study in Serra da Cabreira, Portugal

Ana C.L. Sá[1], Bruno Aparicio[1], Akli Benali[1], Chiara Bruni[1], Michele Salis[2], Fábio Silva[3], Martinho Marta-Almeida[4], Susana Pereira[5], Alfredo Rocha[5], José Pereira[1]

[1]School of Agriculture, University of Lisbon, Lisbon, 1349-017, Portugal
[2]Consiglio Nazionale delle Ricerche, Istituto per la BioEconomia, Sassari, Italy
[3]Autoridade Nacional de Emergência e Proteção Civil, Lisbon, Portugal
[4]Centro Oceanográfico de A Coruña, Instituto Español de Oceanografía, A Coruña, Spain
[5]CESAM - Department of Physics, University of Aveiro, Aveiro, Portugal

*Correspondence to*: Ana C.L. Sá (anasa30@gmail.com)

**Abstract.** This study aims to assess wildfire hazard in northern Portugal by combining landscape-scale wildfire spread modelling and connectivity analysis to help fuel management planning. We used the Minimum Travel Time (MTT) algorithm to run simulations under extreme (95th percentile) fire weather conditions. We assessed wildfire hazard through burn probability, fire size, conditional flame length and fire potential index wildfire descriptors. Simulated fireline intensity (FLI) using historical fire weather conditions were used to build landscape networks and assess the impact of weather severity in landscape wildfire connectivity (DICW). Our results showed that 27 % of the study area is likely to experience high-intensity fires and 51 % of it is susceptible to spread fires larger than 1,000 ha. Furthermore, the increase in weather severity led to the increase in the extent of high-intensity fires and highly connected fuel patches, covering about 13 % of the landscape in the most severe weather. Shrublands and pine forests are the main contributors for the spread of these fires, and highly connected patches were mapped. These are candidates for targeted fuel treatments. This study contributes to improve future fuel treatment planning by integrating wildfire connectivity in wildfire management planning of fire-prone Mediterranean landscapes.



# 1 Introduction

In the last decades, wildfires have had growing economic, environmental, and human losses impacts as a result of changes in climate and land use in the Mediterranean Basin, despite increasing in suppression efforts (Bowman et al., 2017; Tedim et al., 2018). Concomitantly, wildfire management policies focused on fire suppression and ignoring ongoing climate change and landscape-scale fuel build-up, have resulted in very severe wildfires (Curt and Frejaville, 2018; Rodrigues et al., 2019). The large number of simultaneous fire ignitions that often burn at high-fire intensities jeopardize the suppression system putting it beyond the limits of extinguishing capacity (Plucinski, 2019). Hence, to tackle the increased frequency of intense and large wildfires requires combining fire suppression and fuel reduction strategies in landscape-level wildfire management plans. Currently, the effectiveness of such plans has been assessed via reduction in burned area extent, rather through limitation of damages and losses (Moreira et al., 2011, 2020). The failure of that objective has raised the need for a paradigm shift in wildfire management practices towards rebalancing between suppression efforts and prevention measures (Ingalsbee, 2017; Moreira et al., 2020; Palaiologou et al., 2020; Wunder et al., 2021).

There is evidence of past profound socio-economic changes that led to the rural exodus in several countries of the Mediterranean basin. In Portugal, since the 1960's the extensive land abandonment and afforestation, have led to a significant decrease in the agricultural and pastoral activities, which resulted in large changes in landscape configuration and composition. These circumstances have promoted the increase in the fuel load, availability and contiguity (Fernandes et al., 2019; Moreira et al., 2020), which associated with unusually severe meteorological conditions, led to the tragic fire season of 2017. This year had a record-breaking of 557,400 ha of burned area, millions of euros in economic losses and a total of 119 fatalities (Castellnou and et. al., 2018; Ribeiro et al., 2020). From then on, fire management has gained relevance and visibility in the public and political discussion, leading to development of a fuel management plan to be implemented from 2020 to 2030, aiming to reduce national-level exposure to wildfires (RCM, 2021).

One of the main challenges to scientists and wildfire managers is to increase landscape heterogeneity by creating interruptions in large, continuous expansions of forests and shrublands. Measures to break landscape connectivity, like interspersing different land use – land cover classes (LULC), reducing fuel load and contiguity (fuel breaks, wide area treatments and prescribed fires, among others), may hinder the hazardousness of the landscape to large and intense wildfires, promoting the change to less fire prone regions (Moreira et al., 2020). In wildfire research, connectivity concepts have been applied, for example to study the relationship between forest connectivity and burnt areas (Martín-Martín et al., 2013); the link between climate change and fuel connectivity (Fletcher et al., 2016; Keeley et al., 2018); the impact of different weather and forest connectivity levels on fire spread (Duane et al., 2021); and the location of the best subset of fuel treatment units that minimize the impact of the worst-case wildfire (Liberatore et al., 2021). Recently, to quantify the influence of the spatial arrangement of fuels in fire spread connectivity, a new connectivity index was developed, which integrates estimated fireline intensity and the effect of wind direction on fuel connectivity (Aparício et al., 2022). Information derived from this wildfire





connectivity index can be useful to prioritize fuel treatment units, and to identify fire suppression opportunities and ultimately define operational tactics.

Fire spread models can estimate fire spread and behavior under different weather conditions and alternative fuel management scenarios, producing information that can be used in support of wildfire management decisions (Finney, 2006). These models have been widely used to assess: wildfire hazard, exposure and risk (Alcasena et al., 2021; Palaiologou et al., 2020; Salis et al., 2013); wildfire transmission (Oliveira et al., 2016; Salis et al., 2021); and impact of fuel treatments (Benali et al., 2021; Salis et al., 2016a, 2018). In Portugal, fire spread simulations have been used at regional and local scales to analyze

the effectiveness of fuel-break treatments and fire risk transmission in the Algarve region (Oliveira et al., 2016); to propose strategic prioritization of fuel treatments over time in commercial eucalypt plantations (Martín et al., 2016); to compare the impact of different landscape levels of fuel treatments on wildfire hazard reduction (Benali et al., 2021), the impact of different intensity levels of forest management have in financial outcome from timber productions (Barreiro et al., 2021), and the fire risk assessment of human settlements affected by large wildfires (Oliveira et al., 2020), all developed in the

Centre of Portugal. Recently, wildfire spread modelling was also used to quantify national wildfire exposure of Portuguese communities and protected areas to large fires, as a response to support national plan of future wildfire risk mitigation (Alcasena et al., 2021). Commonly, wildfire hazard assessments are based on a set of fire spread descriptors used to locate the most fire prone areas, and hence identify where fuel management actions ought to be implemented, given pre-defined objectives (e.g., lower intensities, smaller burned areas, etc.). Fuel reduction strategies decrease the intensity of fires and can

also create opportunities for wildfire suppression, ultimately leading to a reduction in exposure and risk to people, infrastructures and of ecosystems and their services (Alcasena et al., 2021; Moudio et al., 2021).

Actual wildfire hazard assessments still ignore the relevance of characterizing wildfire connectivity and of identifying the main fuel patches responsible for the spread of intense fires in the landscape. We propose to address this research gap by combining fire spread simulation with landscape connectivity analysis in a study area located in north-western Portugal.

Specifically, our study aims to: 1) assess the landscape wildfire hazard under extreme weather conditions; 2) characterize landscape wildfire connectivity; and 3) identify landscape fuel patches where treatments can be most effective in breaking the connectivity of intense and large fires. Results can be used to enrich the information used in wildfire hazard assessment and to help fuel management planning in other fire-prone Mediterranean landscapes.

## 2 Data and Methods

### 2.1 Study area

The study area (ca. 200,000 ha) is in the north-western Portugal and is centred at Serra da Cabreira ("goat-herder mountain", in Portuguese). The terrain is rugged, with its highest peak at 1262 m of altitude (Fig. 1). Vegetation is adapted to heat and relative dryness, but the influence of both factors is decreased by the regular presence of moist and fresh air masses that come from the Atlantic Ocean (Costa et al., 1998). The combination of abundant Winter precipitation with dry, warm

Summers influences the distribution and composition of the vegetation communities in the mountain. In this north-western pyro-region where there are two annual peaks of fire activity: one relatively small, centred in March, associated with pastoral burning, and the main one in August. The Summer fire season typically extends from July to September (Calheiros et al., 2020).

The main land cover classes are shrublands (25 %), maritime pine forests (17 %), oaks and other hardwood forests (15 %) as 95 extracted from the last national LULC of 2018 (Direção-Geral do Território (DGT), 2021). Agriculture covers approximately 16 % of the area mainly in the valleys (Fig. 1c). Most of the agricultural areas and eucalypt plantations (11 %) are located southwest of the study area, near the interface with urban area. The largest continuous patch of pine forest (ca. 7,600 ha) is located at the eastern limit of the mountain, and it is divided to the north by a patch that burned in 2010. In Serra da Cabreira, there are herds of wild horses, cattle, and goats. Fire is traditionally used for disposing of agricultural stubble and for pasture 100 renewal, which are important causes of ignition within its boundaries and to the southwest of the study area (ICNF, n.d.).

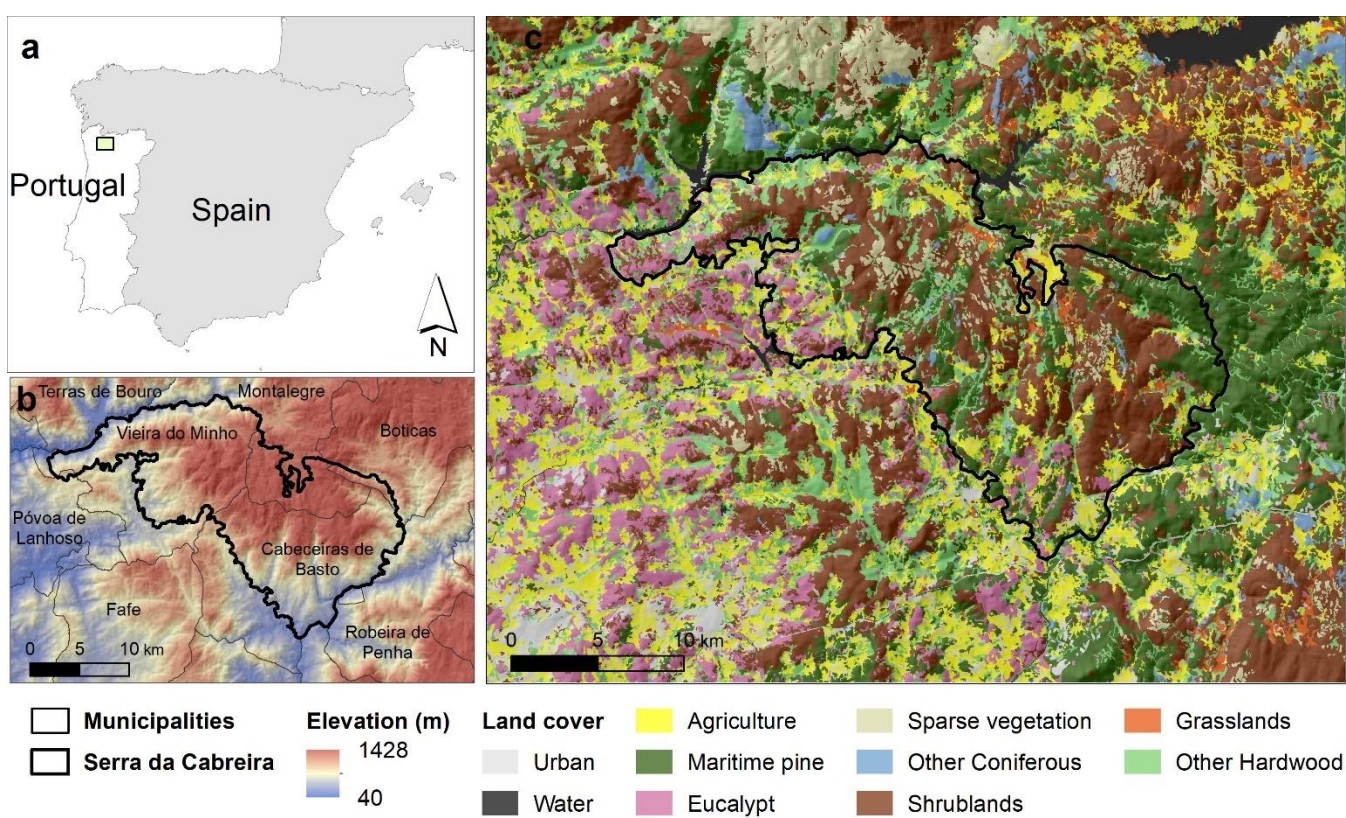

**Figure 1** Study area centred in Serra da Cabreira and its relative position to the Iberian Peninsula (a); Elevation and Portuguese municipalities (b); and main land use / land cover classes.



## 2.2 Fire history


Historic wildfire data for the study area were extracted from the national fire-atlas with 46 years of burned area perimeters (ICNF, n.d.; Oliveira et al., 2012). We selected fires larger than 100 ha that occurred between 2001 and 2019, corresponding to a total of 200 burned area perimeters, which accounted for 64 % of the total burned area in this period. In this subset, there are nine fires (5 %) larger than 1,000 ha, contributing to ca. 25 % of the area burned; and ca. 85 % of the burned area

perimeters have less than 500 ha (Fig. 2d). The largest burned area is 4,300 ha, located at the eastern edge of the mountain (Fig. 2b).

We extracted fire ignition locations from the Portuguese Forest Service fire database, with the start and end dates of the corresponding fires. Different sources of errors may affect the accuracy of this database, such as incorrect location, data loss or misplacement, etc. (Pereira et al., 2011). We used satellite data to complement and improve the accuracy of the location of

ignitions and individual duration following previously developed research (Benali et al., 2016b). Based on this information, we calculated the frequency distribution of fire durations for the analysed 200 fire perimeters.

An ignition probability surface was produced by interpolating the ignition points using an inverse distance weighting algorithm with a fixed radius of 6 km, corresponding to the peak distance above which spatial clustering of ignitions decreases (Fig. 2a). The highest probability of ignition is located to the south of the study area and in two distinct regions of

the mountain range. Fires were historically more frequent in the southern region (Fig. 2b) but the largest fires occurred mainly in the central and eastern regions, where the probability of ignition is lower (Fig. 2c).

**Figure 2** Historic wildfire data description for burned area perimeters larger than 100 ha, from 2001 to 2019: probability of
ignition (a); frequency of burning (b); fire sizes (c); and percentage of the number of fire perimeters and burned area
perimeters by classes of area (d).

## 2.3 Fire weather

Weather variables of temperature (T), relative humidity (RH), wind speed (WS) and wind direction (WD), were compiled for
the spread days of the 200 wildfires. Weather data were estimated from simulations of the Weather Research and Forecast
model (Skamarock et al., 2008). The regional model is based on the configuration described and validated by Marta-Almeida
et al. (2016) and has a spatial resolution of 5 km. Temperature and RH were extracted at 2 m, and WS and WD at 10 m
above the surface, both with 3 hour frequency. Each weather variable results from daily average for the period 12h-20h,





because it commonly represents the hotter, windier part of the day when fire spread is faster and more intense. A summary of
the distribution of average daily values of the selected fire weather variables is shown in Appendix A.

Fire weather data were classified into clusters where centroids represent daily averaged values of T, RH and WS. We used a
model-based clustering classification (Stahl and Sallis, 2012) where each cluster obtained was assigned a weather type.
Details of the clustering method and assumptions are shown in Appendix B.

Table 1 shows the three classified weather types: 1) "frequent/hotter (H)": the most frequent fire weather, which has the
highest mean T; 2) "drier/windier (DWi)": the second most frequent fire weather, which has the lowest mean RH and the
highest mean WS; and 3) "cooler/wetter (CWe)": the least frequent fire weather corresponding to the lowest T and highest
RH values. The latter weather type is associated with wildfires occurring outside the regular fire season, or under milder
weather conditions often observed during the final stages of fire spread. The most frequent wind directions are from
Northeast (41%) and West (20%). The remaining directions have frequencies lower than 10% and are, in descending order,
from East, South, North, Northwest, Southwest and Southeast.

The 95th percentile of T and WS, and the 5th percentile of RH were calculated to characterize an extreme weather condition.
This corresponds to T of 30°C, RH of 24% and WS of 22 km.h$^{-1}$. The frequencies of WD in this subset of days were: 36.5 %
East, 36.5 % South; 18 % Northeast and 9 % North.

Hence, we defined two weather conditions: 1) *historical*, characterized by the three weather types; and 2) *extreme*,
corresponding to the 95th percentile of the fire weather dataset. The first was used to calibrate the fire spread simulation
system and to obtain a reference fire spread simulation. The latter was used to simulate hypothetical large and intense
wildfires.

**Table 1.** Weather types obtained from the classification of the 326 fire weather days (N_days) according to the average
values of Temperature (T), Relative Humidity (RH), and Wind Speed (WS). For each weather type (with frequency Fr), the
percentage distribution of wind direction (WD) was calculated.

| Weather type | N_days | Fr | T (°C) | RH (%) | WS (km/h) | WD (%) | | | | | | | |
|---|---|---|---|---|---|---|---|---|---|---|---|---|---|
| | | | | | | N | NE | E | SE | S | SW | W | NW |
| H | 190 | 0.58 | 25.6 | 43.5 | 10.4 | **10.1** | **28.1** | **11.2** | 1.1 | 9.0 | 6.7 | **27.0** | 6.7 |
| DWi | 105 | 0.32 | 24.6 | 30.3 | 14.4 | 7.5 | **60.4** | 3.8 | 5.7 | 9.4 | 1.9 | 7.5 | 3.8 |
| CWe | 31 | 0.10 | 14.2 | 63.7 | 10.4 | 0.0 | **50.0** | **25.0** | 0.0 | 0.0 | 0.0 | **16.7** | 8.3 |





### 2.4 Fire spread simulation system

#### 2.4.1 FlamMap

We performed fire spread simulations using the Minimum Travel Time (MTT) fire growth algorithm as implemented in the
*FlamMap* simulation system (Finney, 2006). The MTT algorithm calculates fire growth by searching for the set of pathways
with minimum spread time among cells in the two-dimensional gridded landscape at an arbitrary user defined spatial
resolution (Finney, 2002). Wildfire spread is predicted using the Rothermel's model (Rothermel, 1972), which estimates fire
descriptors in the direction of the maximum rate of spread. This algorithm has been used in several fire-prone areas
worldwide to address different wildfire management objectives (Alcasena et al., 2021; Palaiologou et al., 2018; Parisien et
al., 2019). In Portugal, it was used to simulate extreme wildfires and evaluate the impact of fuel treatments in decreasing
landscape wildfire hazard and risk (Benali et al., 2021; Oliveira et al., 2020) and exposure of communities and protected
areas to large wildfires (Alcasena et al., 2021).

#### 2.4.2 Input data

The fire spread simulation system requires a set of input data that includes spatial grid layers to describe the landscape, a list
of fire ignition locations, and information about fire weather conditions and corresponding fuel moisture contents. We
compiled fire weather data (T, RH, WS and WD), fire regime descriptors (burnt area, ignition locations, and corresponding
fire sizes and durations), vegetation (tree cover and surface fuels) and elevation data, to characterize the landscape and its
fire regime. Elevation was obtained from the 30 m Space Shuttle Radar Topography Mission (SRTM, Acker et al., 2014)),
and the corresponding grids of slope and aspect were derived and resampled to 100 m. Vegetation and topography data were
assembled in a common geographic 100-m resolution grid.
The surface fuel model map for 2020 was derived by assigning the Portuguese (Fernandes et al., 2009) and American
(Anderson, 1982) fuel models typologies to the national LULC classes of 2018 and updated in recently burned areas. Tree
cover density for 2018 was downloaded from the pan-European High Resolution Layers in the Copernicus Land Monitoring
Service (European Environment Agency (EEA), 2018). The historic wildfire ignition probability grid (Fig. 2a) was used to
randomly sort the simulated fires.
Temperature and RH were used to calculate the initial values of fuel moisture content (1-h, 10-h and 100-h time-lag dead
fuels classes) using available equations from the literature (Anderson et al., 2015; Nelson Jr, 2000). The herbaceous and
woody live fuel moisture contents were set equal to 60% and 90%, respectively. The WD prevailing distribution frequencies
were those from the described historical and extreme weather conditions.

#### 2.4.3 Simulation settings

Fire modelling was conducted at 100 m resolution using the landscape input data and considering temporally constant
weather and fuel moisture conditions throughout the simulation time. We estimated wildfire descriptors for the historical and



extreme fire weather conditions, and the fuel model grid for 2020. The landscape was saturated with 100,000 fires randomly sampled using the historic probability of ignition and unburnable fuels mask extracted from the fuel model grid. Simulation

spread durations and corresponding frequencies were those obtained from model calibration (Sect. 2.4.4): 300 min. (60 %), 540 min. (25 %) and 780 min. (15 %). Fire suppression efforts and crown fires were not simulated.

### 2.4.4 Calibration

We calibrated the fire spread simulation system using the historic wildfires (burned area, ignitions and durations) larger than 100 ha from 2001 to 2019, the fire weather conditions (weather types) and fuel model grids derived from the Portuguese

LULC maps of 1995 and 2010 (Direção-Geral do Território (DGT, 2021) representative of historical vegetation cover of the study area. The two fuel maps, three fire weather types, and three fire duration classes were combined in a calibration matrix corresponding to each variable combination of frequencies. The two fuel model maps were assigned a frequency according to the total burned area before and after 2010; each weather type frequency was obtained from the model-based classification; and initial fire durations and corresponding frequencies were obtained from the wildfire database.

This calibration matrix was then used to set the number of random fire ignitions used in each simulation run (Appendix C). We sampled a total number of 100,000 random fires using the historical ignition probability and the fuel model maps (to exclude ignitions located in non-burnable areas). We calibrated the fire spread modelling system by running the MTT algorithm for each combination of variables in the calibration matrix, adjusting the duration of fire spread until obtaining a satisfactory reproduction of the historical fire frequency distribution.

The capability of the fire simulation system to reproduce historical fire pattern in the study area was assessed by comparing a set of the descriptors: (i) observed vs. estimated fire size frequency distributions; (ii) estimated burn probability vs. observed fire incidence in the historical period (2001-2019); and (iii) simulated vs. reference burned perimeters for historical wildfires larger than 1000 ha (9 fires), for which the Sørensen's similarity index (Sørensen, 1948) was calculated.

### 2.5 Fire hazard

### 2.5.1 Wildfire descriptors

We analysed simulated fireline intensity (FLI, henceforth fire intensity) and burn probability (BP), and frequency distributions of flame length (FL) and fire size (FS). Fire intensity has a relationship with flame length (FL, m) based on the Byram´s equation (Byram, 1959):

$$FL = 0.0775 \times FLI^{0.46} \,, \tag{1}$$

The MTT algorithm estimates FL distribution from multiple fires burning each pixel, from which the conditional flame length (CFL, m) can be calculated as follows:

$$CFL = \sum_{i=1}^{20}(FLP_i)(FL_i) \,, \tag{2}$$





where $FLP_i$ is the flame length probability of a fire at the $i$th flame length class, and $FL_i$ is the midpoint of each of the 20 $i$th classes of 0.5 m flame length. CFL represents the probability weighted flame length given a fire occurs, and has been used as a proxy for fire hazard (Alcasena et al., 2021; Salis et al., 2013).

Burn probability represents the likelihood that a grid cell will burn considering the total number of simulated fires. It is calculated as:

$$BP_p = \left( \frac{F_p}{N_p} \right), \tag{3}$$

where $F_p$ represents the number of times a pixel $p$ burns and $N_p$ is the number of simulated ignitions. The BP has been routinely used to assess wildfire hazard, exposure and risk, usefull for supporting wildfire and forest management plans (e.g. Benali et al. (2021); Lozano et al. (2017); Salis et al. (2013)).

The FS is a list of ignition points with geographical coordinates and burned area extents. These points were interpolated using an inverse distance weighted algorithm to produce a grid of the expected FS. The combination of FS with the historical ignition surface (IP) was then used to map the fire potential index (FPI) as:

$$FPI = FS \times IP \tag{4}$$

where high values of FPI indicate high likelihood of fire ignitions growing into large fires. Understanding how FPI changes with distance from urban areas can be used to strategically protect villages or infrastructures from fire or implement preventive fuel reduction measures.

Previous wildfire descriptors were compared for the extreme and historical (reference) weather conditions, to assess the increase in wildfire hazard with the increase in the weather severity. Furthermore, for the extreme weather condition, we also combined the estimated BP and FLI, to identify areas most likely to be affected by high intensity fires, which can also be used as a proxy of wildfire hazard. FLI was reclassified into four classes according to its relationship with fire suppression difficulty (Alexander and Cruz, 2019, Appendix D) and BP was divided into quartiles. For simplicity of writing, high intensity and very high intensity fires will henceforth be indiscriminately referred to as high-intensity or high-FLI fires.

Lastly, we analysed how the estimated fire hazard descriptors and changes in FPI with distance to urban areas, relate with the main fire affected land cover types.

### 2.5.2 Wildfire connectivity

The spatial configuration of fuel patches need to be considered in fuel and wildfire management planning, since fuel connectivity influences fire spread, fire size and fire intensity (Duane et al., 2021; Fernandes et al., 2014). A new metric to assess wildfire connectivity was recently proposed (Aparício et al., 2022): the Directional Index of Wildfire Connectivity (DIWC). This metric calculates the connectivity of fuel patches using simulated fire intensities and wind direction as the main driver of fire spread direction. It is calculated as follows:



$$DIWC = \frac{\sum_{i=1}^{n} \sum_{j=1}^{n} (a_i \times FLI_i) \times (a_j \times FLI_j) \times W_{ij}}{A_L^2 \times FLI_{max}^2} \quad (5)$$

where, $a_i$, $a_j$, $FLI_i$ and $FLI_j$ are the area and fireline intensity in patches $i$ and $j$, respectively. $A_L$ is the total landscape extent, and $FLI_{max}$ is the maximum FLI patch value in the study area. The weight matrix $W$ is defined as 1-|sin(α)| or 1-|cos(α)| depending on wind direction quadrant. $W_{ij}$ is 1 when the neighbouring nodes are aligned with wind direction. Distinct fuel patches were created by combining fuel model assigned to each land cover class with the main aspect directions and slope classes derived from its influence on fire spread rate (Butler et al., 2007)

We analysed the impact of weather conditions in landscape wildfire connectivity by calculating the DIWC with FLI simulated for the historical and extreme weather conditions. Then, we used DIWC to map the contribution of each fuel patch to the landscape wildfire connectivity. The relationship between estimated FS and CFL, and wildfire connectivity was also explored. Furthermore, we analysed how the estimated fire hazard descriptors relate with the main land cover types burned. We also assessed the changes of FPI with distance to urban areas as an indicator of the exposure of population to fire hazard.

## 3 Results

### 3.1 Wildfire hazard

We used the calibrated fire spread modelling system (calibration results are shown in Appendix E) to assess wildfire hazard in the study area, under extreme weather conditions. Fig. 3 shows the estimated distributions of the wildfire descriptors BP, FS, CFL and FPI, which values are compared with those from historical simulations (Fig. 4).

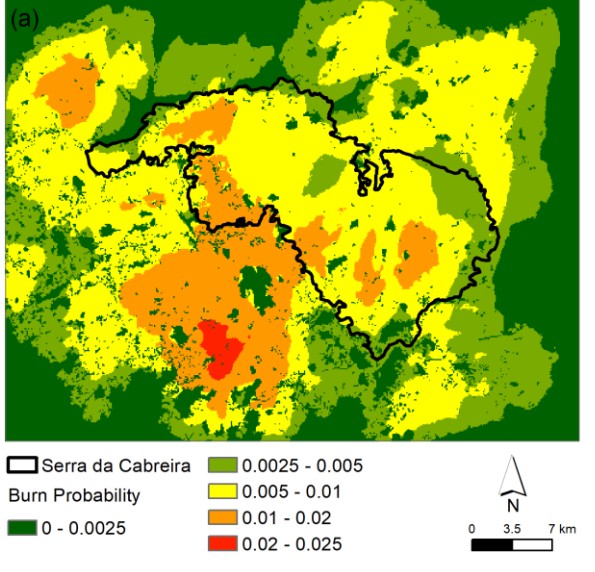

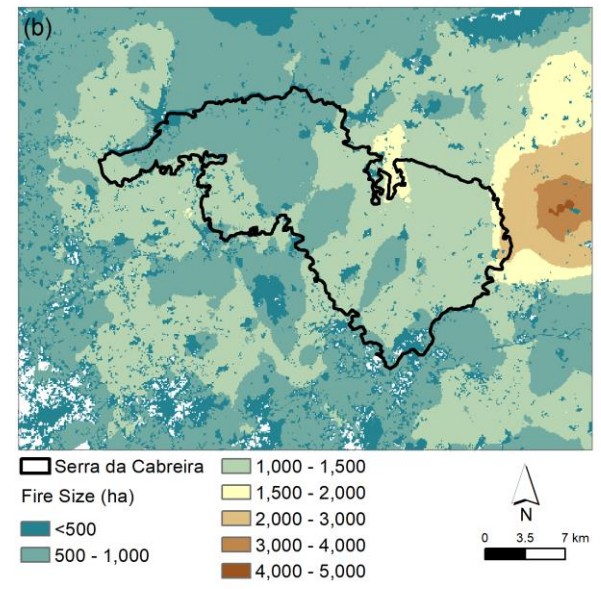


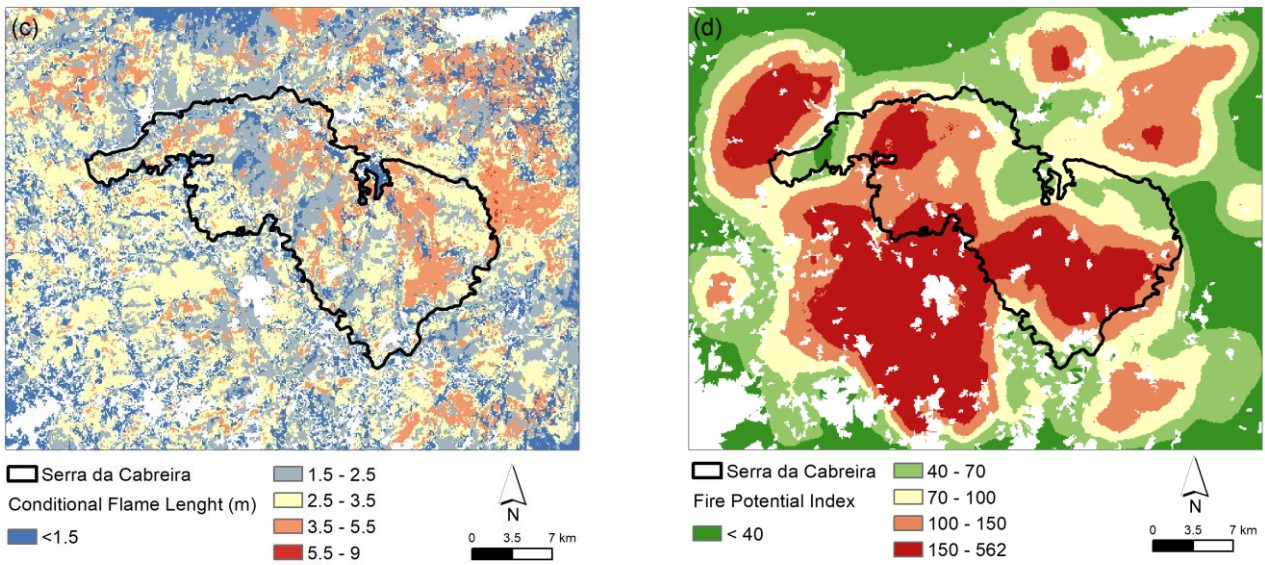

**Figure 3** Wildfire descriptors estimated with the extreme weather fire spread simulations: a) burn probability (BP); b) fire
size (FS); c) conditional flame length (CFL); and d) fire potential index (FPI). FPI is shown in 20th percentile classes.

The highest burn probabilities are in the southwest and northwest of the study area, while the largest sizes are estimated in
the east (Fig. 3a, 3b). Mean simulated BP, FS, CFL and FPI are 0.006, 1,095 ha, 2.5 m and 103, respectively. Comparing the
extreme and historic simulations (Fig. 4a, 4b), on average, BP doubled, and mean FS increased from 461 ha to 1,095 ha (138
%), with 51 % of the study area having fires larger than 1,000 ha.

Approximately 50 % of the study area has CFL values longer than 2.5 m (Fig. 3c), which represent fire intensities that do not
permit suppression at the fire front. Extensive values of CFL longer than 3.5 m are estimated in ca. 15 % of the study area,
mainly in the East (pine forests), and in shrubland areas located in the Northwest and within the mountain limits of the study
area. The spatial pattern of the FPI (Fig. 3d) extends that of the BP, especially in a large part of the southern-central section
of the study area. This likelihood of large fires is higher than the historical mean in approximately 80 % of the study area
(Fig. 3d, 4d).

(a)                                                                 (b)





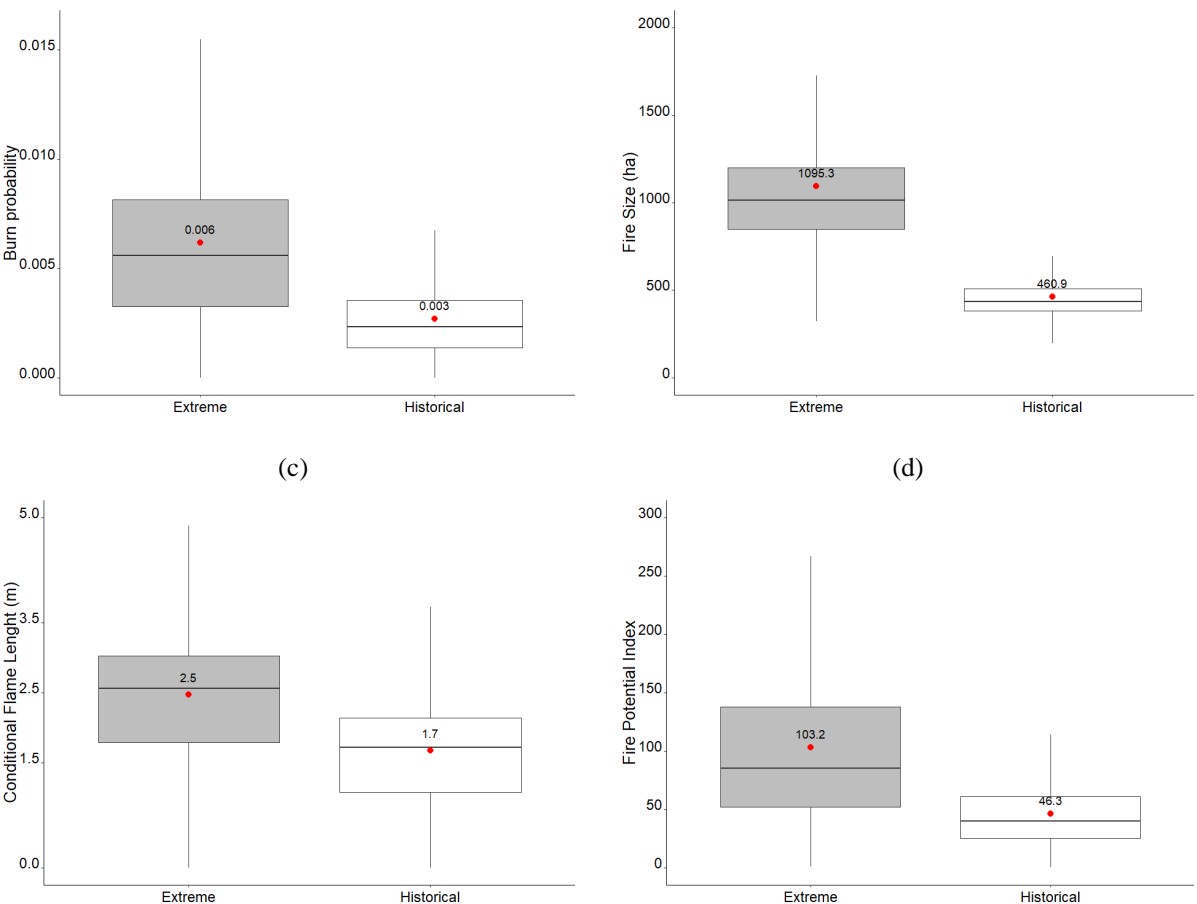

(c)                                                              (d)

**Figure 4** Distribution of extreme and historical estimated wildfire descriptors: a) burn probability (BP); b) fire size (FS); c) conditional flame length (CFL); and d) fire potential index (FPI) for the extreme and historical weather conditions. Red points represent averaged values.

280

By combining the simulated FLI with BP (Fig. 5), the map highlights areas more likely to have intensive fires. Approximately 50 % of the study area has estimated FLI above 4,000 kW m$^{-1}$, which represents areas where suppression is ineffective at the head of the fire, fire spotting and crowing are frequent, and ground-based suppression must be complemented by aerial attack. The most likely locations (BP higher than the median) that spread fires with those intensities cover 27 % of the study area. Also relevant are the areas where, despite being unlikely, fires can spread with high intensity. This represents 25.1% of the area with a BP below the median.

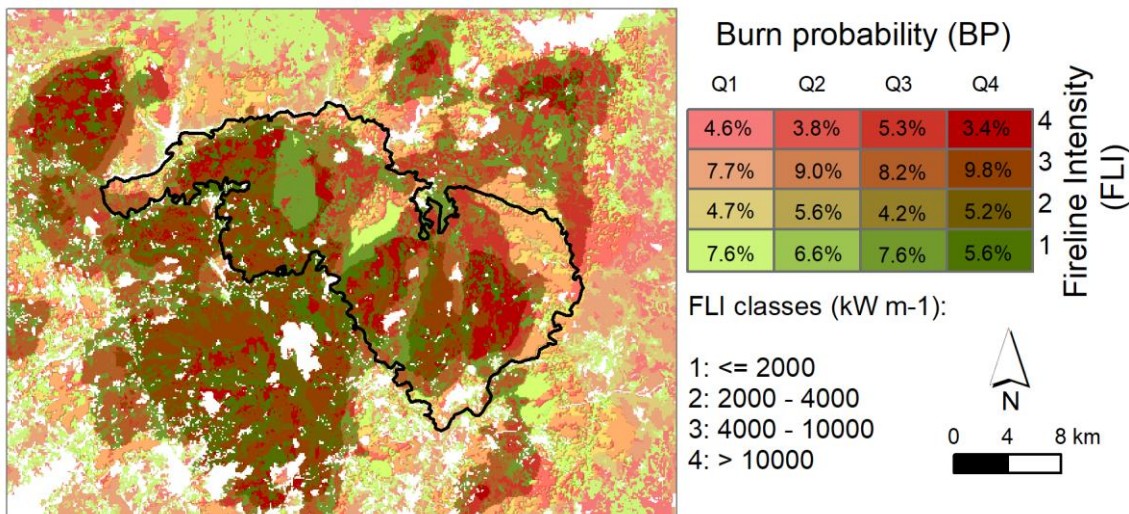

**Figure 5** Spatial combination of the simulated BP and FLI, using burn probability quartiles and FLI classes defined according to fire suppression difficulty, as a proxy of wildfire hazard. In the map lighter colours have lower BP while reddish colours are assigned to higher estimates of FLI.

## 3.2 Land cover hazard

We assessed the relative contribution of different land cover types to the spread of wildfires under extreme weather conditions by calculating their averaged values in the space defined by the BP, CFL, FS and FPI fire spread descriptors (Fig. 6). The main land cover types burned are, in descending order, shrublands, sparse vegetation, other hardwood forests, agriculture, pine forest, grasslands, other coniferous and eucalypt plantations.

Shrublands, eucalypt and other hardwood forests show the largest average BP, representing approximately 50 % of the simulated burned area. However, eucalypts cover a very small fraction (1%) of the burned area, despite their large FPI values. Fires in shrublands and pine forests are expected to be the most intense (CFL of 3.1 m and 2.8 m, respectively) and large (958 ha and 1229 ha, respectively). Fires in grasslands have the largest FS (1260 ha), moderate BP (0.006) and relatively high FPI (41.4). However, only 3 % of the simulated burned area was in grasslands. Other hardwoods represent 18 % of the area burned, with relatively high average BP (0.006), moderate CFL (2.3 m), relatively large FS (826 ha), and intermediate values of FPI (36.5).

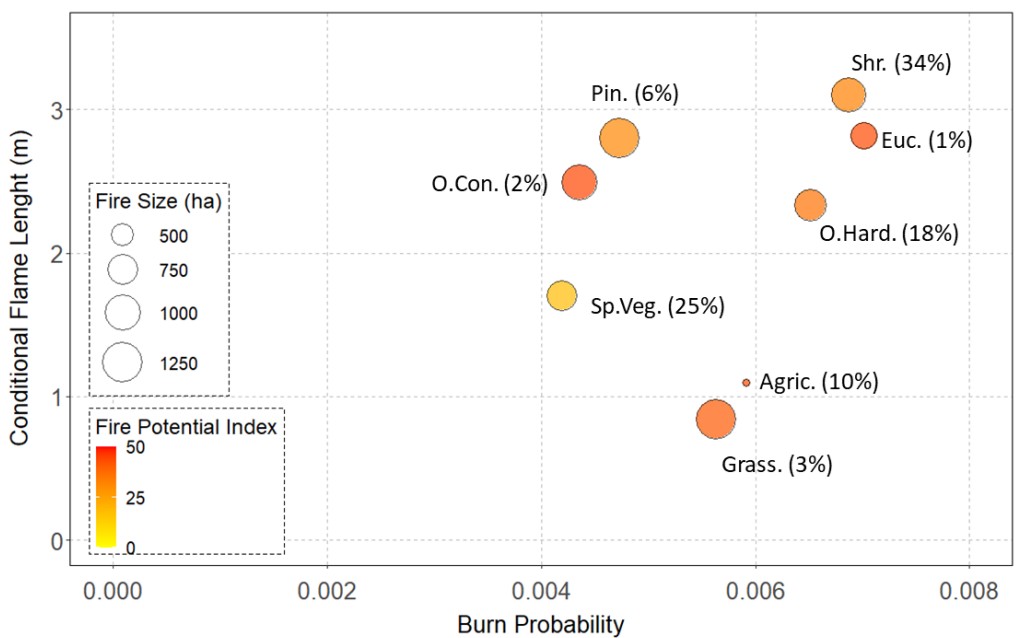

**Figure 6** Averaged values of simulated burn probability (BP), conditional flame length (CFL), fire size (FS) and fire
potential index (FPI) for the eight main landcover classes in the study area. The percentage of each class burned is shown in
parenthesis. Bubble size is proportional to FS, and the colour represents the FPI values. Agric. = Agriculture; Pin. =
Maritime pine; Euc. = Eucalypt; Sp. Veg. = Sparse Vegetation; O.Con. = Other Coniferous; Shr. = Shrublands; Grass. =
Grasslands; O. Hard. = Other Hardwood.

We analysed how FPI changes with distance to urban areas and assessed where the most hazardous land cover classes are
located. (Fig. 7). Comparing with the FPI 75th percentile (Fig. 7a), there is a clear increase in the probability of an ignition
becoming a large fire for distances up to 1 km from urban areas. Up to 250 m from urban areas, agricultural areas and
eucalypt plantations are the most represented classes. Shrublands (38 %), pine (20 %), and other hardwood forest (18 %)
contribute to the large values of FPI, mainly between 500 m and 1 km from urban areas (Fig. 7b). For distances between 1
km and 4 km, the FPI decreases with the increase in sparse vegetation, while pine forests and shrublands decrease. The
lowest FPI values were estimated for distances larger than 4 km from urban areas, where ca. 75 % of the burned area is in
sparsely vegetated locations.



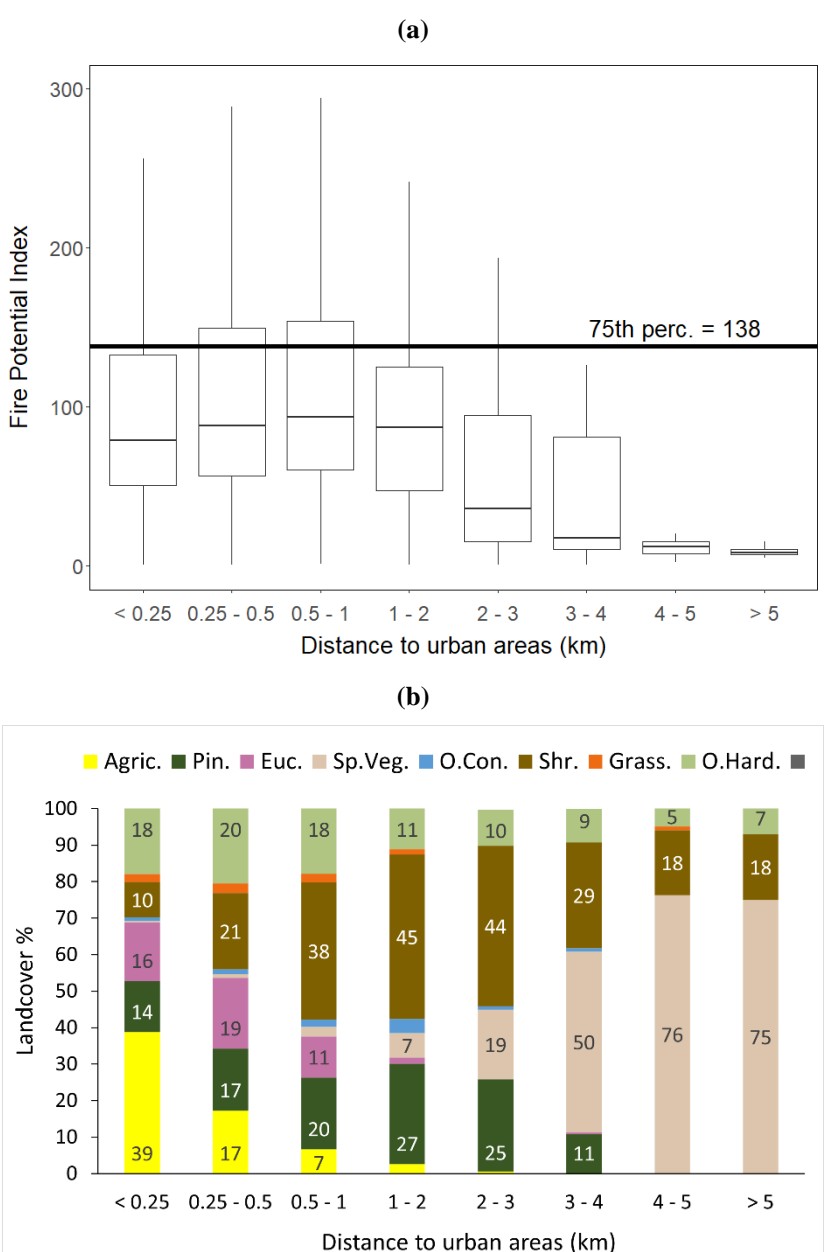

**Figure 7** Fire potential index (FPI) as a function of distance from urban areas (a); and main land cover percentages by
distance from urban areas class (b). Only land cover classes that burned more than 5 % are shown.



### 3.3 Wildfire connectivity

We analysed variations in landscape connectivity for the spread of fires as a function of the fire weather conditions (Fig. 8). With increasing weather severity, the area of the landscape that spread high intensity fires increases. This is clearly shown in the DWi weather type where the DIWC approximately doubles (ca. 0.3) the connectivity of the other weather types; and in

the extreme weather where different WD result in different landscape DIWC values. The highest value of wildfire connectivity was estimated for the North-South wind directions.



**Figure 8** Percentage of fire intensity (FLI) classes and normalized wildfire connectivity index (DIWC) for each weather

condition. Simulated FLI was classified in five classes (Appendix D). Yellow boxes represent the weather types from the historical weather condition while the red box represents the extreme weather condition. Acronyms refer to the weather types and the corresponding wind directions with frequency higher than 10 %.

Figure 9 shows the expansion of hight-FLI classes (from 9 % under CWe to 50 % under P95) with the increase in fire

weather severity. This leads to the coalescence of fuel patches, which the DIWC quantifies as an increase in wildfire

connectivity, as shown by the expansion of mainly two hotspots in the eastern and central regions of the study area. Highly connected patches (with DIWC > 0.10, here selected as having values larger than the 95th percentile) represent 13.3 % (13,125 ha) of the area under the extreme weather, 12 % and 8 % under the DWi and H weather types, respectively. For the extreme weather, these patches are shrublands (57 %), pine forests (22 %) and eucalypt plantations (12 %), where pine

forests (followed by the shrublands) have the largest DIWC values, for all the weather conditions (Appendix F).

**Figure 9** Normalized wildfire connectivity (DIWC) calculated for the historical: CWe (a), H (b), and DWi (c); and extreme (d) fire weather conditions. Only fuel patches with estimated FLI > 4000 kW m$^{-1}$ (high-FLI classes).




Furthermore, Fig. 10 shows for the extreme weather condition and high-FLI patches (Fig. 9d), the relationship between wildfire descriptors and wildfire connectivity values. The fuel patches with the highest values of DIWC also have higher values of FS and CFL (Fig. 10a, 10b). Thus, the location of fuel patches with extreme values of DIWC in the landscape highlights areas likely to spread very intense and large wildfires (median values of 3.7 m and 1,010 ha, respectively). Nonetheless, there is not a relationship between DIWC and BP (Fig. 10c) and patches with higher DIWC have lower FPI

(Fig. 10d), because they are in areas of low probability of fire ignition.

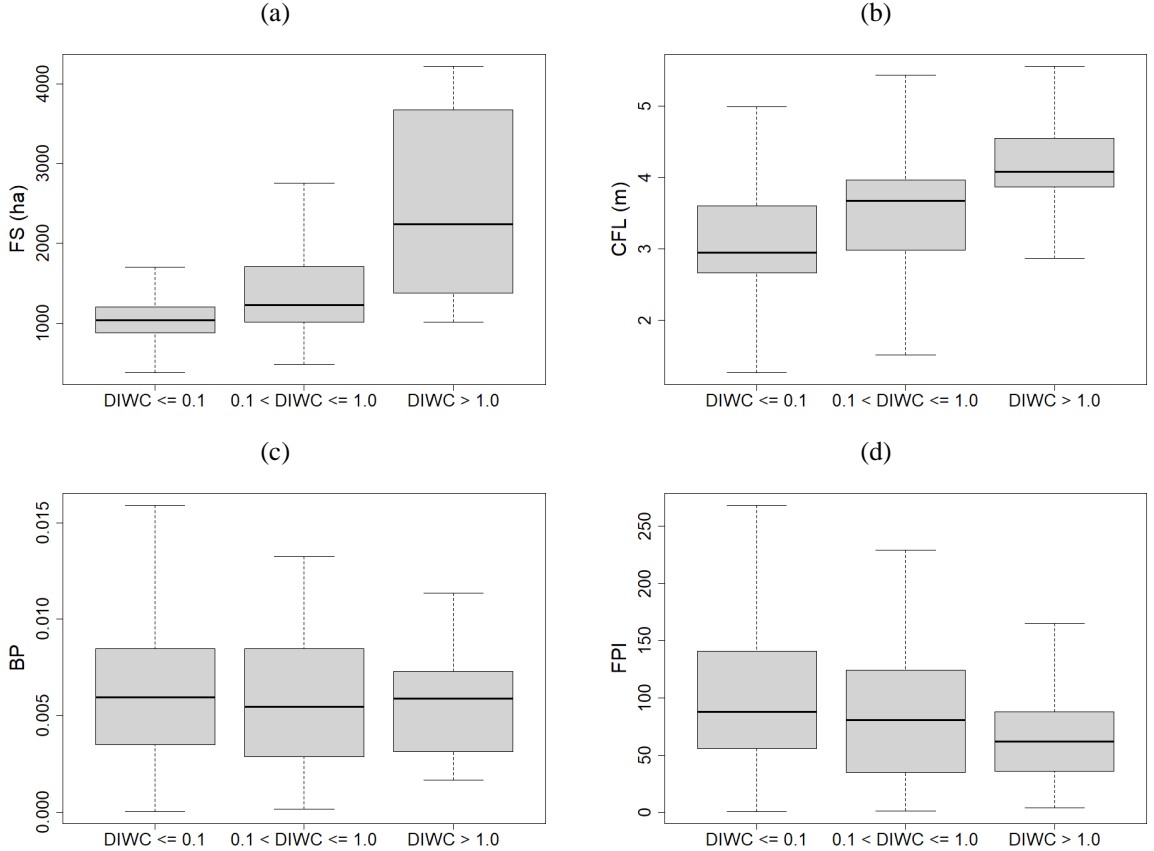

**Figure 10** Relationship between simulated fire size (FS), conditional flame length (CFL), burn probability (BP) and fire potential index (FPI) with the normalized wildfire connectivity index (DIWC), under the extreme weather condition. The 95th percentile of the DIWC is 0.10.





**4 Discussion**

In Mediterranean countries there is an urgent need to adapt fire preventive measures and bring together researchers, politicians, and managers to tackle the prospective increase of wildfire impacts in a changing climate (European Commission, 2021). This requires a paradigm shift that assumes coexisting with fire by creating fire-resilient landscapes. Hence, wildfire management needs to evolve towards identifying the best treatment opportunities that reduce fire intensity
and burned extent, while simultaneously creating opportunities for more effective suppression efforts (Curt and Frejaville, 2018; Wunder et al., 2021).

Consistent with previous findings, our study showed that by combining wildfire hazard and wildfire connectivity assessments supported by fire spread simulations, it is possible to enrich information used in landscape fuel management planning. We located the most likely areas to burn, and those that spread large (above 1,000 ha) and intense wildfires (above
4,000 kW m$^{-1}$). We also showed how landscape wildfire connectivity increases with weather severity and identified fuel patches that mostly contribute to the spread of high intensity fires.

Wildfire hazard assessment under extreme weather conditions showed that Serra da Cabreira is exposed to large and intense fires that mostly spread with eastern and southern winds. Historical fire regime indicates that East of the mountain, the probability of ignition is low. However, simulations showed that the potential largest fires are located here, in extensive
patches of pine forest with high fuel loads. This area has several highly connected fuel patches that support the spread of intense fires into the mountain, likely to burn extensive shrubland areas.

Another important hazardous area extends from the south to the centre of the study area, where intensive and large fires are also expected to spread over shrublands. Southernly, the landscape is more anthropic, where urban areas are interspersed with agricultural lands, eucalypt plantations, and other hardwoods in the valleys. However, at high altitudes this
heterogeneous vegetation pattern is replaced by continuous areas of shrublands where fire frequency is high and fire return intervals can be lower than 5 years, which are both related with the frequent use of fire as a tool for pasture renewal (Catry et al., 2009; Moreira et al., 2011). This cultural use of fire is an important source of fire ignitions in the region, which in hot and windy days may lead to fire hazard increase in the mountain. Another relevant hotspot that has high probability of spreading intensive and large fires is located northwest in another high-altitude shrubland area. Although western winds are
not coincident with extreme fire weather condition, they should not be ignored, given their moderate frequency in climate type H and the possibility that they lead to the spread of forest fires from this area to the mountain.

The maximum potential of likely ignitions spread to large fires lies between 500 m and 1,000 m from urban areas, where shrublands prevail. Here, fuel-load reductions should be planned to decrease wildfire hazard, increase landscape fire-resilience, and improve wildfire response system. With other objectives in mind, such as for example to decrease the impact
of fires in the wildland-urban interface, the FPI hazard descriptor should be replaced by other variable (e.g. fireline intensity), and smaller buffer distances from urban areas have to be considered (Calkin et al., 2014).



Extensive areas of fuels are one of the major determinants of fire size in Portugal (Duguy et al., 2007; Fernandes et al., 2016), thus measures promoting the disruption of fuels contiguity will inevitably create opportunities for fire suppression, decrease burned areas and consequently landscape hazardous. To implement those measures, landscape connectivity
assessment is crucial to fuel treatment planning, but in the perspective of fuel structural connectivity (Liberatore et al., 2021; Rachmawati et al., 2016). Ignoring the complex dynamic fuel-weather interactions, which result in different landscape fire spread patterns, can lead to underestimation of connectivity and even different solutions of where to prioritize fuel treatments (Duane et al., 2021; Zeller et al., 2020).

To address the previous research gap, we applied a recently developed connectivity metric (Aparício et al., 2022) to calculate
and map landscape wildfire connectivity response to the increase in fire weather severity. With increasing fire weather severity, the landscape extent that potentially spreads high-intensity fires increases, and with coalescence of fuel patches, the landscape is more connected to the spread of large and intense fires. In the extreme weather condition, 50 % of the landscape can support the spread of high-intensity fires. In these areas, high wildfire impacts and suppression difficulties are expected., which can be exacerbated by the highly connected to the landscape fuel patches. These are candidate locations for fuel
management treatment aiming to disrupt fire spread connectivity. Nonetheless, the effectiveness of breaking fuels connectivity to mitigate fire impacts and the spread of large fires in the landscape may be significantly reduced by the occurrence of severe weather conditions (Duane et al., 2021). However, there is evidence that there is a fuel effect on fire behaviour under less severe weather conditions (e.g. Anderson et al., 2015). Besides this, as fuel treatments significantly reduce fuel load, it is expected that they significantly decrease fire intensity and the impacts in the landscape, while
simultaneously improving effectiveness of fire suppression operations. Furthermore, those extreme weather conditions typically occur in few days of the fire season, so fuel treatments still have an important role in reducing fire spread and intensity in more frequent less severe weather conditions. We believe that wildfire managers can still use our study results and framework to target fuel patches for treatments aiming at decreasing landscape wildfire impacts.

Although the calibration of the fire spread modelling system reproduced reasonably well the historical fire size distribution
and burned area pattern in the study area, there are some limitations to the current study. Uncertainty in fuel model assignment to existent land cover maps have important impacts on simulation results (Benali et al., 2016a), thus local information should be used to refine fuel models input map. Furthermore, crown fires were not simulated because of the absence of data describing canopy fuels, which may be overcome in the next future by using estimates from LiDAR data (ICNF, 2021). Regarding the surface fuel models, the recently available national land cover map (DGT, 2021) with its
increased spatial resolution, potentially provides more accurate land cover mapping, which may improve the quality of fuel model assignment.



## 5 Conclusions

This study provides research-based information to enrich landscape wildfire management decisions by integrating wildfire connectivity analysis and simulated wildfire hazard descriptors. For the extreme weather conditions, we located the most

hazardous areas of large and intense fires, and we showed that shrublands and pine forests are the land cover types that mostly burned in those damaging fires. We also showed that landscape wildfire connectivity increases with fire weather severity, because of the coalescence of severe fires that extended for ca. 50 % of the study area. Landscape wildfire connectivity was mapped for each fire weather condition, highlighting fuel patches where the potential to spread large and severe wildfires is high.

We believe that these results can help fire managers to identify hot-spot areas where site-specific fuel treatment operations should be planned. Ultimately, they contribute to mitigate future wildfire impacts and increase landscape fire-resilience of Mediterranean fire-prone regions. Future work should include the wildfire connectivity metric in the design of alternative fuel treatment scenarios to inform more sustainable and effective wildfire management in fire-prone Mediterranean landscapes.






# 6 Appendices

## Appendix A. Summary of fire weather data

There are four days of fire spread with T< 10 °C, which correspond to fires that occurred outside the main fire season (Fig.
A1a). Most of the wildfires spread with T between 30 and 35 °C. Most of the days have RH of 40 %, and a quarter below 30
%. In general, lower T are related with higher RH, which often corresponds to the last hours of fire spread, sometimes
coincident with a decrease in severity of the fire weather conditions (Fig. A1b). The most frequent WS lie between 10 and 15
km h$^{-1}$, while the maximum value is 25 km h$^{-1}$. The most frequent WD are from East (41 %) and West (19 %), while each of
the remaining directions has a frequency below 10 % (Fig. A1c, d).

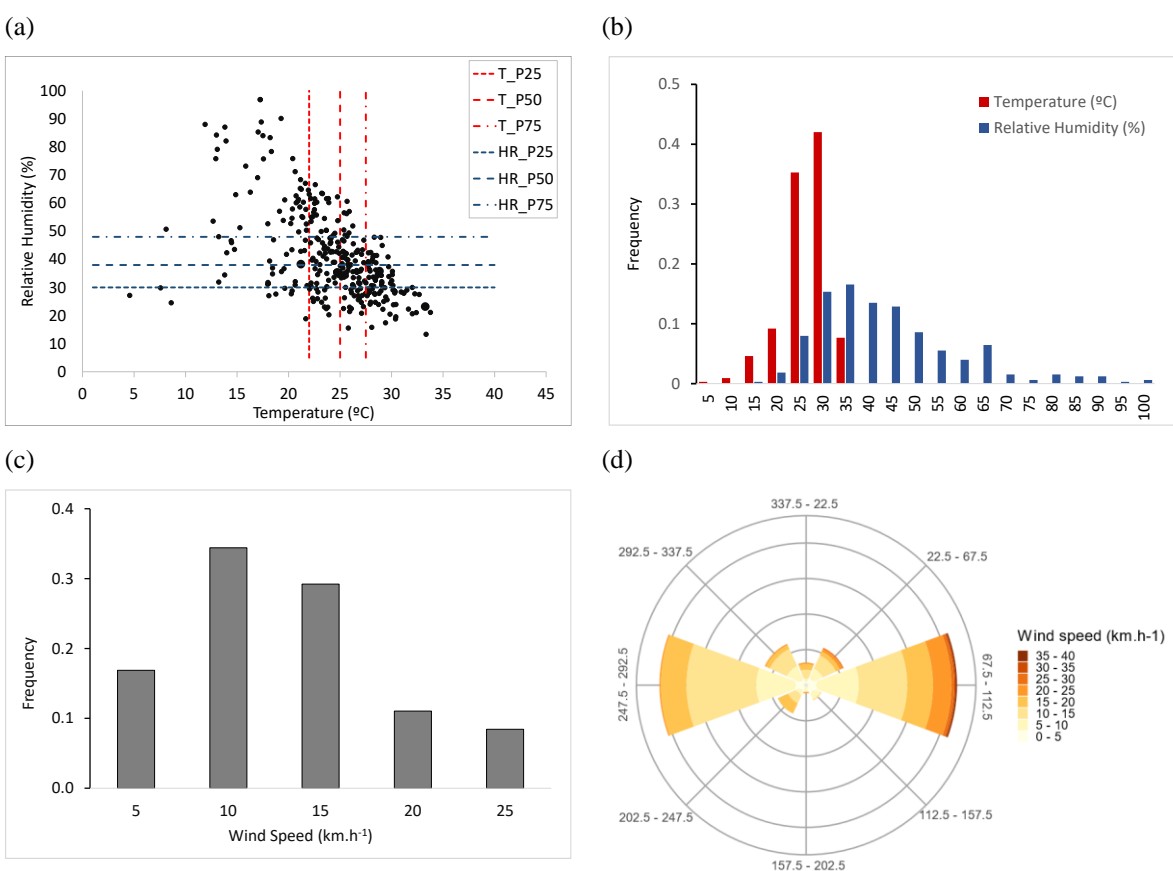

**Figure A1** Fire weather variables compiled for the days of fire spread of the 200 wildfires from the fire database. Average
daily values of the variables were calculated for: temperature and relative humidity, with quartile lines shown (a); frequency
distribution of T and RH (b); frequency distribution of WS (c); and wind rose (d). A total number of 326 days were
compiled.



**Appendix B: Model-based cluster analysis**

A finite mixture model was fitted to the wildfire weather database using the Bayesian Information Criterion model selection to derive the optimal number of clusters. This model considers the data as coming from a distribution that is mixture of two or more cluster. In the classification, the optimal number of clusters is calculated automatically; it integrates uncertainty in class assignment and produces the probability of each daily observation belonging to each cluster. It also produces the geometric features (orientation, size, and shape) of the clusters (Banfield and Raftery, 1993).

Results showed that the optimal solution is a 3-clusters model (Fig. B1a), which have ellipsoidal shapes with varying volume, shape, and orientation (VVV). Uncertainty of cluster allocation is shown in Fig. B1b, where larger symbols indicate more uncertainty. Observations in the left cluster (cluster 2) are more certain of being in the correct cluster, while observations classified in the right clusters are more uncertain as they are more similar.

(a)                                                      (b)

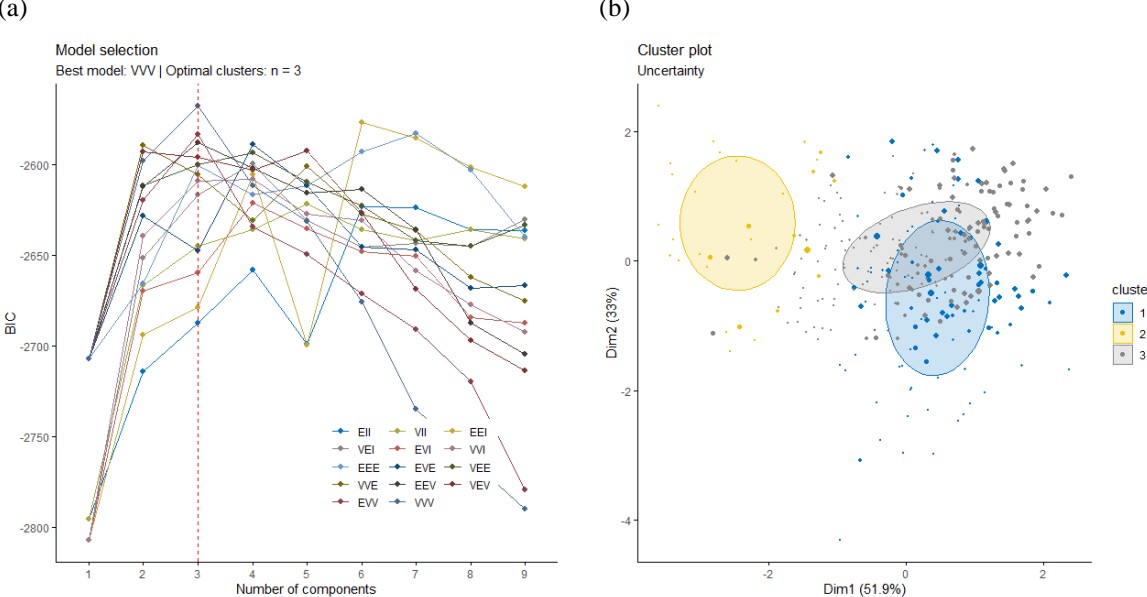

**Figure B1** Optimal solution (three clusters) according to the ellipsoidal volume, shape and orientation properties (a); and uncertainty in cluster assignment of each weather day (b).




**Appendix C: Calibration of the fire spread modelling system**

Calibration was carried out by combining historical fire and weather data regimes in different simulation scenarios with
probabilities corresponding to selected variable frequencies. The resultant calibration matrix (Table C4) was then used to
obtain the number of fire ignitions in each simulation scenario. We run 120 fire spread simulations and a total of 100,000 fire
ignitions. Variables used in the calibration were: 1) two fuel model maps (1995 and 2010); 2) three weather types; 3) wind
direction frequencies in each weather type; and 4) three classes of fire spread duration (table shows final durations used in
the calibrated model).

**Table C1** Calibration matrix defining the fire spread simulation runs and corresponding probabilities.

|  | Weather |  | WindDir | Freq. | Number of IGNITIONS | | | |
|---|---|---|---|---|---|---|---|---|
|  |  |  |  |  | DURATION \| min (freq.) | | | |
|  |  |  |  |  | 300 (0.60) | 540 (0.25) | 780 (0.15) | TOTAL |
| cos1995 (0.4) | 1 | 0.32 | N | 0.08 | 580 | 242 | 145 | 966 |
|  |  |  | NE | 0.60 | 4637 | 1932 | 1159 | 7728 |
|  |  |  | E | 0.04 | 290 | 121 | 72 | 483 |
|  |  |  | SE | 0.06 | 435 | 181 | 109 | 725 |
|  |  |  | S | 0.09 | 725 | 302 | 181 | 1208 |
|  |  |  | SO | 0.02 | 145 | 60 | 36 | 242 |
|  |  |  | O | 0.08 | 580 | 242 | 145 | 966 |
|  |  |  | NO | 0.04 | 290 | 121 | 72 | 483 |
|  | 2 | 0.10 | N | 0.00 | 0 | 0 | 0 | 0 |
|  |  |  | NE | 0.50 | 1200 | 500 | 300 | 2000 |
|  |  |  | E | 0.25 | 600 | 250 | 150 | 1000 |
|  |  |  | SE | 0.00 | 0 | 0 | 0 | 0 |
|  |  |  | S | 0.00 | 0 | 0 | 0 | 0 |
|  |  |  | SO | 0.00 | 0 | 0 | 0 | 0 |
|  |  |  | O | 0.17 | 400 | 167 | 100 | 667 |
|  |  |  | NO | 0.08 | 200 | 83 | 50 | 333 |
|  | 3 | 0.58 | N | 0.10 | 1408 | 587 | 352 | 2346 |
|  |  |  | NE | 0.28 | 3910 | 1629 | 978 | 6517 |
|  |  |  | E | 0.11 | 1564 | 652 | 391 | 2607 |
|  |  |  | SE | 0.01 | 156 | 65 | 39 | 261 |
|  |  |  | S | 0.09 | 1251 | 521 | 313 | 2085 |
|  |  |  | SO | 0.07 | 938 | 391 | 235 | 1564 |
|  |  |  | O | 0.27 | 3754 | 1564 | 938 | 6256 |
|  |  |  | NO | 0.07 | 938 | 391 | 235 | 1564 |
|  |  |  |  |  |  |  |  | 40000 |

|  | Weather |  | WindDir | Freq. | DURATION \| min (freq.) | | | |
|---|---|---|---|---|---|---|---|---|
|  |  |  |  |  | 300 (0.60) | 540 (0.25) | 780 (0.15) | TOTAL |
| cos2010 (0.6) | 1 | 0.32 | N | 0.08 | 869 | 362 | 217 | 1449 |
|  |  |  | NE | 0.60 | 6955 | 2898 | 1739 | 11592 |
|  |  |  | E | 0.04 | 435 | 181 | 109 | 725 |
|  |  |  | SE | 0.06 | 652 | 272 | 163 | 1087 |
|  |  |  | S | 0.09 | 1087 | 453 | 272 | 1811 |
|  |  |  | SO | 0.02 | 217 | 91 | 54 | 362 |
|  |  |  | O | 0.08 | 869 | 362 | 217 | 1449 |
|  |  |  | NO | 0.04 | 435 | 181 | 109 | 725 |
|  | 2 | 0.10 | N | 0.00 | 0 | 0 | 0 | 0 |
|  |  |  | NE | 0.50 | 1800 | 750 | 450 | 3000 |
|  |  |  | E | 0.25 | 900 | 375 | 225 | 1500 |
|  |  |  | SE | 0.00 | 0 | 0 | 0 | 0 |
|  |  |  | S | 0.00 | 0 | 0 | 0 | 0 |
|  |  |  | SO | 0.00 | 0 | 0 | 0 | 0 |
|  |  |  | O | 0.17 | 600 | 250 | 150 | 1000 |
|  |  |  | NO | 0.08 | 300 | 125 | 75 | 500 |
|  | 3 | 0.58 | N | 0.10 | 2111 | 880 | 528 | 3519 |
|  |  |  | NE | 0.28 | 5865 | 2444 | 1466 | 9775 |
|  |  |  | E | 0.11 | 2346 | 978 | 587 | 3910 |
|  |  |  | SE | 0.01 | 235 | 98 | 59 | 391 |
|  |  |  | S | 0.09 | 1877 | 782 | 469 | 3128 |
|  |  |  | SO | 0.07 | 1408 | 587 | 352 | 2346 |
|  |  |  | O | 0.27 | 5631 | 2346 | 1408 | 9384 |
|  |  |  | NO | 0.07 | 1408 | 587 | 352 | 2346 |
|  |  |  |  |  |  |  |  | 60000 |



**Appendix D**

**Table D1** Classes of fireline intensity (FLI) and flame length (FL) translated to fire suppression difficulty. Adapted from
Alexander and Cruz (2019).

| Class | FL (m) | FLI (kW m⁻¹) | Fire Suppression Difficulty |
|-------|--------|--------------|------------------------------|
| 1 | < 1.5 | **< 500** **(very low)** | Fire can generally be attacked at the head or flanks using hand tools. |
| 2 | 1.5 - 2.5 | **500 – 2,000** **(low)** | Fires are too intense for direct attack on the head using hand tools. Equipment such as plows, dozers, pumpers, and retardant aircraft can be effective in suppression |
| 3 | 2.5 - 3.5 | **2,000 – 4,000** **(moderate)** | Fires may present serious control problems – torching out, crowning, and spotting. Control efforts at the fire head will probably be ineffective |
| 4 | 3.5 - 5.5 | **4,000 – 10,000** **(high)** | Crowning, spotting, and major fire runs are frequent. Control efforts at head of fire are ineffective. Aircrafts are required for fire suppression |
| 5 | > 5.5 | **> 10,000** **(very high)** | Any combat attempt (even with aircrafts) is ineffective |






**Appendix E. Calibration of the fire spread modelling system**

The uncertainty in fuel data and meteorology (Benali et al., 2017), duration of fire spread and the lack of knowledge of the
conditions that drove the spread of each individual fire, led to the development of a calibration framework based in an array
of weights, used to generate fire ignitions and to weight the output simulated wildfire descriptors (Table C1). We covered the
historic period of analysis as the combination of frequencies covering two vegetation fuel maps, three fire weather types,
wind distribution frequency in each of these weather types and three fire spread durations. The calibrated model was
obtained by tunning fire durations until the distribution of the simulated fires described reasonably well the historical fire
patterns. We ended the calibrated model with the next fire durations and corresponding frequencies: 300 min (60%); 540 min
(25%); and 720 min (15%).

Figure E1a shows that using this combination of durations, we reproduced reasonably well the historical fire size distribution
pattern. The simulated and reference burned area perimeters peak at ca. 200 ha but with a clear underestimation of the
number of simulated fires in this class. The opposite occurs for burned areas between 500 ha and 1500 ha, where there is an
overestimation of the frequency of simulated fires. One of the reasons for this might be that fire suppression is not
considered in the simulations. The estimated BP map (Fig. E2) reproduces very well the spatial pattern of the frequency of
burn in the study area between 2001 and 2019 (Fig. E1b). Additionally, the highest burn probability regions are coincident
with those that historically had higher ignition probability (Fig. 2a), and that burned more frequently (Fig. 2b).


(a)         (b)

**Figure E1** Comparison between the simulated and observed burned area (a); and between the estimated burn probability and
the historical frequency of burning (b).




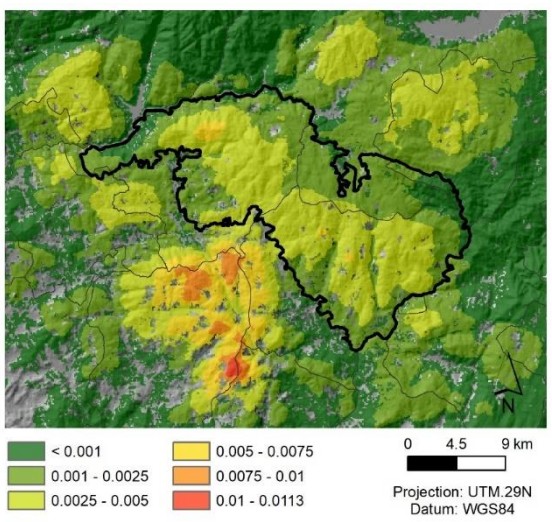

**Figure E2** Simulated burn probability (BP) derived from running fire spread simulations using 100,000 random ignitions
with fire spread durations of 5h (60%), 9h (25%) and 13h (15%). Relative frequencies are shown in brackets.

Moreover, with the calibrated fire modelling system we simulated the spread of the 9 largest wildfires (> 1000 ha, and
responsible for approximately 25% of the total burned area between 2001 and 2019 in the study area) using the
corresponding fire weather data, and the duration of 13h. The Sørensen's similarity index was 0.60, in the interval limit
between moderate and substantial agreement classes (Filippi et al., 2014). This values is in agreement with values obtained
in other fire spread simulations (Alcasena et al., 2016; Salis et al., 2016b). Overall, these results show that the calibrated
wildfire modelling system accurately reproduces the historical size and spatial distribution of fires in the period of analysis.


**Appendix F.**

The landcover classes that have FLI estimates larger than 4,000 kW m-[1] are shrubs (57%), pine forests (22%) and eucalypt plantations (12%). The remaining classes cover less than 10% with those FLI values. Wildfire connectivity for each simulated fire weather condition is shown in Fig. F1. Pine forests have the largest wildfire connectivity from all the weather conditions, showing similar values for P95 and DWi. Eucalypt plantations have the lowest values of DIWC in all weather conditions.


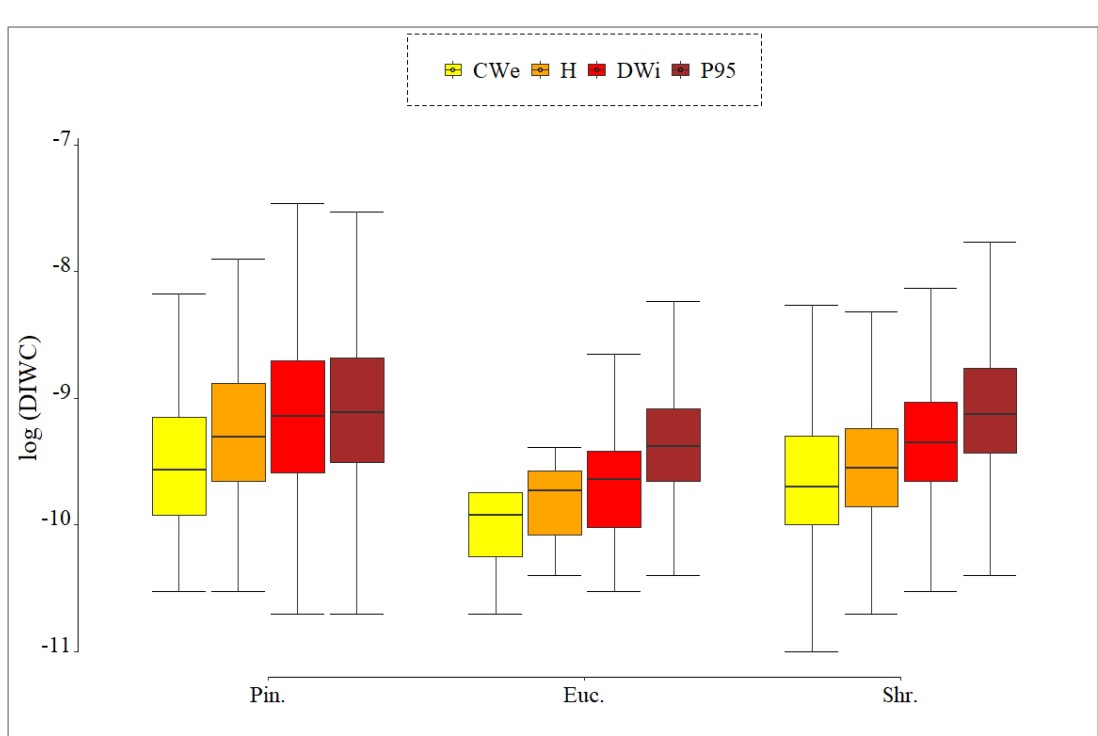

**Figure F1** Natural logarithm of the normalized wildfire connectivity (DIWC) of patches with FLI above 4,000 kW m-[1], estimated from fire spread simulations with historic (CWe, H, DWi) and extreme (P95) weather conditions. Pin. = pine forest; Euc. = eucalypt plantations; Shr. = shrublands.




**Data availability**

All raw data can be provided by the corresponding authors upon request.

**Author contributions**

AS, JP, BA, AB, MS and FS were involved in the conceptualization; AS, MS md JP developed the methodology; AR supervised SP and MMA in the weather data processing; CB explored the fire spread modelling system and prepared input weather data; BA performed the connectivity analysis; AS and BA analysed the data; BA and AS developed R-codes for 540 data analysis and visualization; AS developed the calibration framework; AS wrote the manuscript draft and edited the co-authors revisions; all co-authors were involved in the review of the manuscript.

**Competing interests**

The authors declare that they have no conflict of interest.

**Acknowledgements**

The authors would like to thank the Portuguese forest pulp companies The Navigator Company and ALTRI Florestal for the shared information of the land cover in the industrial properties, used to derive the national fuel model map of 2020, under the framework of the FIRE-MODSAT II project. We also thank Fabio Silva and his colleagues Pedro Machado e Carlos Branco from the Portuguese Special Force of Civil Protection for their fully availability and interest in collaborating with fieldwork related with fuel model assignment in the Serra da Cabreira.

**Funding**

This research was funded by the Forest Research Centre research unit, funded by Fundação para a Ciência e a Tecnologia I.P. (FCT), Portugal (UIDB/00239/2020). B.A. Aparício was supported by the Ph.D. fellowship funded by FCT (UI/BD/150755/2020). A.C.L. Sá was supported under the framework of the contract-program nr.1382 (DL 57/2016/CP1382/CT0003). A. Benali was supported by the research contract (CEECIND/03799/2018/CP1563/CT0003). C. 555 Bruni grant was supported by the project foRESTER, a project funded by FCT (PCIF/SSI/0102/2017). J.M.C. Pereira was supported by the FireCast - Forecasting fire probability and characteristics for a habitable pyroenvironment project (PCIF/GRF/0204/ 2017), funded by FCT. The participation of S. Pereira and A. Rocha was supported by the project FIRE-MODSAT II (PTDC/ASP-SIL/28771/2017) also funded by FCT.



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
