# Peer review of "Coupling wildfire spread simulations and connectivity analysis for hazard assessment: a case study in Serra da Cabreira, Portugal"

_Natural Hazards and Earth System Sciences, 2022_

## Author Comment (AC1)

We would like to thank the reviewer´s suggestion that is relevant to compare fireline intensity (as measured by the CFL estimate) with the forest fire loss (FFL) dataset, particularly for the year of 2021, as the first fire season after the wildfire hazard assessment. However, we need to remind some considerations regarding the fire regime in the study area and the framework of the fire spread simulations, which make them not totally comparable with the FFL dataset:

- Fuel model map used in the simulations includes burned areas from 2020, thus one approach would be to compare it with forest fire loss map of 2021 (FFL21);
- However, in 2021 the largest fires have 118 and 124 ha, and are both winter fires (from January and March, respectively); According to the FFL21 none of these fires were highlighted with forest loss;
- Wildfire hazard was assessed using the 95$^{th}$ percentile of fire spread days to represent the extreme weather conditions that usually can produce large fires, specifically >= 100 ha;
- The fire regime in the study area is characterized for having some winter fires (represented in the "cooler/wetter (CWe)" weather type) that spread with weather conditions different from those occurring in the main fire season and in extreme days;
- According to the official Portuguese Statistics, there are some discrepancies between these and those from the FFL dataset; Annual burned area values are not supposed to be lower than the area identified as Forest Loss due to fires. As we can see in the histogram below (Figure 1), there are three years where this occurs: 2008, 2014 and 2018 (in the last case the difference is ca. 1600 ha). This makes us suspect that at least in these years there might be a problem with the FFL dataset;
- From the previous point, we do not have sufficient and comparable fire data from 2021 fire season in the study area.

Besides these considerations, and assuming that fire simulations were run at 100m spatial resolution (lower than the 30m from the FFL dataset) under extreme weather conditions and ignoring the lack of correspondence between the observed burned area extent with the corresponding annual records from the FFL dataset, we compared the CFL estimates with the annual area of FFL from 2001 to 2021.

We compared the FFL divided into two periods (before and after 2010) with the estimated CFL divided in two classes (below and above 2.5m, based on the relationship with FLI). We removed from analysis the years 2008, 2014 and 2018 due to the reasons mentioned above.

The hypothesis is that areas where in the past were classified with FFL are expected to have higher intensities now, especially if they are not from recent fires; For this reason, we selected the year of 2010 to define two burned area periods that historically burned approximately the same.

Results are shown in the next table:

| FFL | CFL (m) | |
|---|---|---|
| | <2.5 | >=2.5 |
| before 2010 | 20.6% | **79.4%** |
| after 2010 | **41.1%** | 58.9% |

From the table above, we conclude that areas that before 2010 were mapped with forest fire loss ("sum of high and medium certainty of forest loss due to fire pixels") are mostly expected to experience very intense fires (~79%). Areas that burned after 2010 and that were mapped with forest fire loss, approximately 60% has the potential to spread high intensity fires, as shown by CFL values >= 2.5 m.

The next table shows the distribution among years before and after 2010, and we can see that before 2010 the burned areas from years of 2006 and 2010 are those that now are more susceptible to have very intense fires. 73% of the areas that burned after 2010 (mainly from 2016 and 2017) are susceptible to less intense fires (CFL<2.5m) due to the lower accumulation of fuels, as expected. Overall, we may conclude that with this validation we believe that our wildfire hazard assessment based on the estimated intensity of fires is reasonably reliable using the Portuguese forest service reference data and the suggested dataset of the Forest Fire Loss

| FFL | CFL | | TOTAL |
| --- | --- | --- | --- |
| | <2.5 | >=2.5 | |
| before 2010 | | | |
| 2001 | 7.1% | 4.7% | 5.2% |
| 2002 | 14.2% | 8.4% | 9.6% |
| 2003 | 0.5% | 0.4% | 0.4% |
| 2004 | 5.6% | 4.6% | 4.8% |
| 2005 | 9.3% | 5.5% | 6.3% |
| 2006 | 25.5% | 22.4% | 23.0% |
| 2007 | 3.5% | 2.5% | 2.7% |
| 2009 | 17.3% | 12.2% | 13.3% |
| 2010 | 17.0% | 39.2% | 34.6% |
| after 2010 | | | |
| 2011 | 2.1% | 6.9% | 5.0% |
| 2012 | 9.4% | 12.3% | 11.1% |
| 2013 | 7.1% | 8.4% | 7.8% |
| 2015 | 4.4% | 10.9% | 8.2% |
| 2016 | 20.6% | 34.0% | 28.5% |
| 2017 | 52.5% | 18.0% | 32.2% |
| 2019 | 1.3% | 3.0% | 2.3% |
| 2020 | 0.4% | 0.6% | 0.5% |
| 2021 | 2.1% | 5.9% | 4.3% |

[Figure]

Figure 1

---

## Author Response (AR1)

**Point-by-point response to reviews**

**Referee #1**

**Minor comments**

1. Many sentences in the manuscript are very long, which makes them hard to follow. Please, review the entire manuscript text and try to break long sentences into smaller pieces.

Reply: We agree with the reviewer, and we revised the manuscript to avoid long sentences throughout it

2. L132-134: According to the authors, WRF output was provided at 3h intervals and the weather variables used were averaged over the 12-20h time window. How can this affect the credibility of their analysis, particularly with respect to extreme fire weather? Are the authors certain that the 3h WRF output and subsequently the averaging allows for pinpointing extreme fire weather? A comment on that, also within the manuscript, would be highly appreciated.

Reply:

We thank the reviewer whose questions are relevant, and we need to clarify them within the simulation framework. There are two main reasons for averaging weather values over the 12-20h time window:

1) We do not have information for individual fire events regarding the active fire spread periods and on the corresponding driving weather conditions. Therefore, we calibrated the fire modelling system using the historical fire regime instead of individual fire-based calibration. Considering this data limitation, we run simulations with constant weather for spreading durations lower than 24h. Therefore, based on expert knowledge, we know that in the study area the main active fire spread frequently occurs during the 12-20h time window, when higher temperature and wind speed, and lower humidity conditions prevail.
2) Historical large fires have spreading durations larger than 3h, and as shown in the calibration matrix and considering that suppression is not being modelled, the most frequent duration in the calibrated model is 5h (60%), which in a real case scenario fire duration is longer.

Impact:

- The averaging effects that may decrease extreme values of the weather variables used will be compensated by the tunning of the duration of simulation, which is the output parameter that is extracted from the calibration and allows the reproduction of the historical fire regime. The duration parameter is not a real duration of a single fire but, different durations are used to cover the full range of fire sizes observed in historical data (Appendix E)
- We believe that we are not failing to estimate the fire spread under extreme weather conditions, as shown in the results of the accuracy assessment done for the simulation of the 9 largest fires (above 1000 ha, usually driven under extreme weather conditions), in Appendix E.

As suggested, we included the next sentence in the final version of the manuscript (Lines 130-134):

*"This time window choice is because fire simulations are run for spread durations smaller than 24h with constant weather conditions. Averaged values of the weather variables needed to exclude milder weather conditions that typically occur during the evening and morning periods, when fire spread is reduced also due to effective fire suppression efforts. The eventual averaging effect of the extreme weather conditions is compensated by tunning the duration in model calibration (Sect. 2.4.4)."*

Technical remarks
L22: improve --> improving
L28: increasing in suppression efforts --> increased suppression efforts
L34: rathern than
L91: remove where (?)
L130: Weather Research and Forecasting model
L465: Table C4 --> Table C1
L466: run --> ran
L511: This value or These values
L409: on the landscape
L418: future instead of next future

All these remarks were corrected in the final version of the manuscript.

This is a well develop research methodology and should be consider for publication in NHESS. To further improve:

My only suggestion would be to include a small analysis of validation (accuracy assessment) of your predicted fire size or intensity/ hazard against forest loss due to fire dataset for Serra da Cabreira in 2018 (https://glad.umd.edu/dataset/Fire_GFL/).

**Reply**:

We would like to thank the reviewer´s suggestion of including a small analysis of accuracy assessment to further improve the study. We did not understand why 2018 was the year selected for comparison since the dataset was updated to include 2020 and 2021.

It is relevant to compare fireline intensity (as measured by the CFL estimate) with the forest fire loss (FFL), particularly for the year of 2021, as the first fire season after the wildfire hazard assessment. We need to highlight some considerations regarding the fire regime in the study area and the simulation framework:

- Fuel model map used in the simulations includes burned areas from 2020, thus we compared it with forest fire loss map of 2021 (FFL21) only including in the analysis those pixels coded with medium and high certainty.
- Wildfire hazard was assessed using the 95th percentile of fire spread days to represent the extreme weather conditions that usually can produce large fires, specifically >= 100 ha
- In 2021 the largest fires have 118 and 124 ha, and are both winter fires (from January and. March, respectively). According to the FFL21 none of these fires were highlighted with forest loss.
- According to the official Portuguese Fire Statistics database, there are some discrepancies when compared to the FFL dataset. Annual burned area values cannot be lower than the area identified as Forest Loss due to fires in 2021. As we can see in the histogram below (Figure 1), there are three years where this is true: 2008, 2014 and 2018 (in the last case the difference is ca. 1600 ha). This raises suspicions that at least in these years there might be a problem with the FFL dataset.

Besides these considerations, and assuming that fire simulations were run at 100m of spatial resolution (lower than the 30m from the FFL dataset) under extreme weather conditions, and ignoring the lack of correspondence between the reference burned area extent with the corresponding annual records of the FFL dataset, even though we compared the CFL estimates with the annual area of FFL from 2001 to 2021.

The hypothesis is that areas that were recorded with FFL more recently, should have lower estimated CFL values, thus being able to generate lower intensity fires. Areas that were in the past classified with FFL are expected to have higher intensities now, especially if they are not from recent fires.

Therefore, we summed the total area of FFL divided in three periods (before 2010; between 2010 and 2016; and after 2016) and quantified the area of the estimated CFL divided in two classes (below and above 2.5m, based on its relationship with FLI). We removed from

analysis the years 2008, 2014 and 2018 due to the reasons mentioned above. Results are shown in table A, and indicate that areas that burned more than 5 years ago were in 2021 able to generate high intensity fires, in more that 70% of the areas that were recorded in the past with FFL (only those with moderate and high confidence). Oppositely, in approximately 60% of the areas that burned after 2016 with assigned FFL, these have a lower potential to generate high intensity fires.

Concluding, with this small analysis of validation, we have some evidence that in the last five years, areas that were recorded with FFL, have in 2021 a lower chance of spreading high intensity fires, as measured by the CFL.

Table 2.

| FFL | CFL (m) | |
|---|---|---|
|  | <2.5 | >=2.5 |
| before 2010 | 21.0% | **79.0%** |
| 2010 - 2016 | 29,7% | **70,3%** |
| after 2016 | **62,4%** | 37,6% |

We included the next paragraphs in the final version of the manuscript, according with the previous results:

- As a simple validation exercise we evaluated if areas that were classified in the past with forest fire loss in the "Global forest loss due to fire" dataset (Tyukavina et al., 2022) are expected to have higher intensities in 2021, as estimated by the CFL. This dataset has a higher spatial resolution and we only used pixels coded with moderate and high certainty in the analysis. Comparison with the Forest Fire Loss (FFL) dataset was done by summing the total area of FFL divided in three periods (before 2010; between 2010 and 2016; and after 2016) and quantifying the area of the estimated CFL divided in two classes (below and above 2.5m, based on its relationship with FLI) (section 2.5.1)

- Comparison between the estimated CFL and the FFL dataset showed that the CFL is likely providing good results in the estimated areas of high intensity fires (CFL>2.5m), thus where it is expected higher fire impacts (Table 2). A large percentage of the areas that had FFL before 2016 (above 70%) are likely to have CFL values above 2.5m, thus high intensity wildfires in 2021. In recent burnt areas (latter than 2016), most of the areas that had FFL (62.4%) are prone to lower intensity fires (section 3.1)

- We also included Table 2 next to the previous paragraph from section 3.1.

Regarding other corrections made in the manuscript we only added some corrections in the reference section.